# Primipaternity in multiparas as a predominant high risk factor for preeclampsia over prolonged birth intervals: A study of 33,000 singleton pregnancies in Reunion Island

Pierre-Yves Robillard[1,2]* , Silvia Iacobelli[1,2‡], Simon Lorrain[2], Francesco Bonsante[1,2‡], Malik Boukerrou[2,3‡], Marco Scioscia[4‡], Phuong Lien Tran[3‡], Gustaaf Dekker[5]

1 Service de Réanimation Néonatale et Pédiatrique, Néonatologie, Centre Hospitalier Universitaire Sud Réunion, Saint-Pierre Cedex, La Réunion, 2 Centre d'Etudes Périnatales Océan Indien (CEPOI, UR 7388), Université de la Réunion, Centre Hospitalier Universitaire Sud Réunion, Saint-Pierre Cedex, La Réunion, 3 Service de Gynécologie et Obstétrique, Centre Hospitalier Universitaire Sud Réunion, Saint-Pierre Cedex, La réunion, 4 Department of Obstetrics and Gynaecology, Mater Dei Hospital, Bari, Italy, 5 Department of Obstetrics & Gynaecology, Robinson Institute, University of Adelaide, Lyell McEwin Hospital, Adelaide, Australia

☯ These authors contributed equally to this work.
‡ These authors also contributed equally to this work
* robillard.reunion@wanadoo.fr, pierre-yves.robillard@chu-reunion.fr

## Abstract

### Objectives

To evaluate the relative importance of changing paternity ("primipaternity", direct inquiry with patients) in multiparas versus prolonged birth/pregnancy interval as risk factors for preeclampsia (PE) by a logistic regression model comparing the adjusted odds ratios of both exposures.

### Design

Assessment of all consecutive singleton deliveries (from 22 weeks onwards) at South-Reunion University's maternity (Reunion Island, Indian Ocean) over 23 years (2001–2023) using an epidemiological perinatal database on obstetrical factors (264 items in total, of which, chronic or gestational hypertension, proteinuria, HELLP syndrome).

### Results

Among the 53,572 multiparous singleton pregnancies, we identified 33,312 (62%) of multiparas who gave consecutive births, allowing calculation of birth intervals.

Primipaternity multipara (N = 2790) were on average older than those in stable relationships (N = 50,782), 31 vs 30 years, p< 0.0001; they had almost systematically longer birth intervals compared with controls of approximately 1.5 year from the 2nd to the 4th pregnancy and approximately 1year after the 5th pregnancy (all p < 0.05). In the logistic regression model of 11 risk factors, intervals between pregnancies had similar adjusted odds ratios

**Data Availability Statement:** All relevant data are within the manuscript and its Supporting information files.

**Funding:** The author(s) received no specific funding for this work.

**Competing interests:** The authors have declared that no competing interests exist.

(1.05, p = 0.002) as increasing maternal age (AdjOR 1.02, p = 0.02), increasing parity (adjOR 1.09, p = 0.02) and pre-pregnancy BMI (AdjOR 1.05, p< 0.0001). Smoking was associated with an AdjOR of 0.85 (non-significant),primipaternity multiparas were twice as likely to be smokers (23.8% vs 13.4%, p< 0.0001) compared with controls. AdjOR for primi-paternity was 3.34 (p < 0.0001) indicating that primipaternity as risk belonged in the category of well-established risk factors like history of preeclampsia (11.2, p< 0.0001) and chronic hypertension (6.45, p< 0.0001).

## Conclusions

Primipaternities in multiparae belongs to the major risk factors such as history of preeclampsia, chronic hypertension, multiple pregnancies while prolonged birth intervals belongs to moderate "regular physiological aging processes" such as increasing maternal age, parity or increasing pre-pregnancy BMI.

## Introduction

In 2002 Skjaerven et al. stated "After adjustment for the interval birth, a change of partner is not associated with an increased risk of preeclampsia" (PE) after analysis of The National Birth Registry of Norway during the period 1967–1998 (on 500,000 multiparas); paternity being probably based on a change of patronymic name of successive siblings [1]. In 2001, one year earlier, Trogstad et al [2] on the exact same population and period of study also concluded that "the impact of changing paternity on preeclampsia risk had been confounded by insufficient control for the time interval between the pregnancies", but they also found that prolonged birth interval is not a risk factor in women with previous PE with or without new paternity.

Interestingly, Trogstadt et al. investigated the effect of prior miscarriage/abortions (< 22 week's gestation) in nulliparous women within a Norwegian MoBa cohort [3]. This study identified that prior to abortion with the same partner a reduced risk of PE was observed however, not in women with new paternity pregnancies. These findings suggested that normal pregnancies interrupted at early gestation may induce immunological changes that reduce the risk of preeclampsia in subsequent pregnancies. Importantly, a study by Skjaerven et al.'s [1] had tremendous impact as it directly contrasted our preceding publications on the importance of 'primipaternity' [4,5].

In those studies we showed that in Guadeloupe (French West Indies) a change of paternity for the index pregnancy, based on direct inquiries with multiparas, was strongly associated with PE [5]. Over the past decades the relative importance of prolonged birth interval versus primipaternity has remained a controversial topic. From an etiological perspective, this ongoing scientific debate is not a trivial one; evidently, the primipaternity paradigm is in line with the fundamental concept that human placentation may be considered as a "fetal hemigraft", and as such the classic superficial cytotrophoblast invasion of the spiral particularly in early-onset PE with fetal growth restriction could therefore represent a type of immunological maladaptation of this fetal hemigraft [6–9]. On the other hand, prolonged interval between pregnancies in multiparas as a direct major risk factor for PE appears to be more in line with a kind of vascular maternal problem that increases progressively with time (a kind of "aging approach"). This was also supported by Tanberg et al. (also in Norway) who concluded on a cohort of 500,000 mothers after assisted reproductive technologies (ART, period 1988 to 2009

–so with a ten year overlap of the prior Skjaerven cohort) that the PE risk may increase by parity, interbirth interval and advanced maternal age, but with not with change of father or smoking [10]. In contrast, in 2000 a study published by Li and Wi based on 140,147 women with two consecutive births (Californian birth certificate 1989–1991) [11] among women without PE/eclampsia in the 1st pregnancy, changing partners resulted in a 30% increase in the risk in the subsequent pregnancy compared with those who did not change partner (95% CI: 1.1–1.6). On the other hand, among women with PE in the 1st birth, changing partner resulted in a 30% reduction in the risk of PE in the subsequent pregnancy (95% CI: 0.4–1.2). Interbirth interval was very unlikely to be a confounder in the Li and Wi study since the authors restricted their population to births that were between 1–3 years apart [11]. Hercus and Dekker in an Australian population studied this problem in 2020 [12] and concluded that "both prolonged birth intervals and primipaternity are independent risk factors for preeclampsia in multigravidae".

Notwithstanding the fact that we have previously discussed some concerns regarding the Skjaerven's study [6,7,13], it is clear that the relative importance of prolonged birth interval versus primipaternity as PE risk factor in multiparous women still represents an important fundamental research question.

The aim of this study was to address this fundamental question by a comprehensive analysis of our 23year pregnancy cohort in Reunion island using a detailed high quality perinatal database where we have an item "changing paternity" by direct inquiry to the women (PE and controls).

## Material and methods

In order to analyse the relative importance of changing paternity ("primipaternity") in multiparas versus prolonged birthinterval as risk factors for PE, we used the exposures "changing paternity", and birth interval both items captured in the database bedides the standard booking characteristics. In multiparous women, the question of possible changed paternity is directly asked to all individual patients at time of the booking visit since 2018. Risk factors for PE between multiparae with a new male partner for the index pregnancy were compared with stable couples.

From January 1$^{st}$, 2001, to December 31, 2023 (23 years), the hospital records of all women who gave birth at the maternity department of the University South Reunion Island were abstracted in a standardized fashion. The study sample was drawn from the hospital perinatal database which prospectively records data of all mother-infant pairs since 2001. Information is collected at the time of delivery and at the infant hospital discharge and regularly audited by appropriately trained staff. This epidemiological perinatal data base contains information on obstetrical risk factors, description of delivery, and maternal and neonatal outcomes. For the purpose of this study, records have been validated and have been used anonymously. All pregnant women in Reunion Island as part of the French National Health Care System have their prenatal visits, biological and ultrasound examinations, at the time of the morphology scan and anthropological characteristics recorded in a maternity booklet. Access to maternity care is free of charge as provided by the French healthcare system, which combines freedom of medical practice with nationwide social security. Hospitals have European standards of care health care, in particular maternity services are based on scheduled appointments (8 prenatal visits and on average 4 ultrasounds).

## Design and study population

The maternity department of Saint Pierre hospital is a tertiary care centre that performs about 4,300 deliveries per year, thus representing about 80% of deliveries of the Southern area of

Reunion Island, and it is the only level-3 maternity (the other maternity is a private clinic only providing low risk maternity care, level 1). Reunion Island is a French overseas region in the Southern Indian Ocean.

Definition of exposure and outcomes. During the 23-year period all consecutive singleton pregnancies after 22 weeks gestation have been analysed. For multiparas, women who had changed the male partner for the index pregnancy (primipaternity) were considered as cases, those who did not were the controls.

Preeclampsia was defined according to the World Health Organization recommendations [14] and the International Society for the study of Hypertension in Pregnancy) relatively to the guidelines in force at the year of pregnancy. [15] as the new onset of hypertension (BP $\geq$140 mmHg systolic or $\geq$90 mm Hg diastolic) at or after 20 weeks' gestation and substantial proteinuria ($> 0.3$ g/24 hours). Early onset preeclampsia was defined as preeclampsia resulting in birth $< 34$ week's gestation.

Data are presented as numbers and proportions (%) for categorical variables and as mean and standard deviation (SD) for continuous ones, as appropriate. Comparisons between groups were performed using $\chi^2$-test and odds ratio (OR) with 95% confidence interval (CI) was also calculated. Paired t-test was used for parametric and the Mann-Whitney $U$ test for non-parametric continuous variables. P-values $<0.05$ were considered statistically significant. Epidemiological data have been recorded and analysed with the software EPI-INFO 7.1.5 (2008, CDC Atlanta, OMS), EPIDATA 3.0. EPIDATA allowing the adaptation for WINDOWS 10 of the former EPI-INFO (MS DOS) in complete cooperation with CDC Atlanta. All calculations were made then with and EPIDATA Analysis V2.2.2.183. Denmark.

To validate the independent association of different outcomes Preeclampsia (PE), early onset PE, Late onset PE. Maternal age and other confounding factors on different sorts of SGA we realized a multiple regression logistic model. Variables associated with all kinds of PE in bivariate analysis, with a p-value below 0.1 or known to be associated with the outcome in the literature were included in the model. A stepwise backward strategy was then applied to obtain the final model. The goodness of fit was assessed using the Hosmer-Lemeshow test. A p-value below 0.05 was considered significant. All analyses were performed using MedCalc software (version 12.3.0; MedCalc Software's, Ostend, Belgium).

We considered the following covariates as possible confounders in this analysis with the outcome Preeclampsia, EOP (early onset), LOP (late onset): maternal age by increment of 5 years of age, pre-pregnancy BMI by increment of 5 kg/m$^2$, smoking during pregnancy, chronic hypertension,. Among multiparas: previous preeclampsia, previous SGA, previous abortion, previous miscarriage, interval between births and "primipaternity". We included these variables and calculated the $\chi^2$ for trend (Mantel extension), the odds ratios for each exposure level compared with the first exposure level.

Ethics approval: This study was conducted in accordance with French legislation. As per new French law applicable to trials involving human subjects (Jardé Act), a specific approval of an ethics committee (comité de protection des personnes- CPP) is not required for this non-interventional study based on retrospective, anonymized data of authorized collections and written patient consent is not needed. Patients and Public involvement. The South-Reunion perinatal database (since 2001) includes 264 items. It is considered as a fully medical database, datasheets are electronically completed solely by midwives, obstetricians and neonatologists. All epidemiological studies are obligatorily performed on anonymized data (French law). As such, there is no direct patient or public involvement.

## Results

During the 23-year period (January 1, 2001- December 31, 2023) there were 86,376 singleton pregnancies. Among the 53,572 multiparous singleton pregnancies, we could identify 62%v of multiparas with consecutive births (33,312/53,572) in our maternity, allowing the calculation of birth intervals.

Table 1, depicts the major maternal characteristics with in the right columns known pre-eclampsia risks in all our singleton multiparas, N = 53,572.

There were 2790 multiparas with a new father for the index pregnancy vs 50,782 stable couples. Primipaternity multiparous women were more prone to be older 31 vs 30 years, to be aged over 35 (OR 1.57), grand multipara (5 children and over, OR 1.5), to have more gravidi-ties (4.3 vs 3.7), parities (2.3 vs 2.0), to live single (OR 4.7), to smoke during pregnancy (OR 2.0), all characteristics p< 0.0001. Overall these women were less educated (10th grade and over), OR 0.88, p = 0.002. There were no differences for mean pre-pregnancy BMIs and rate of obesities ($\geq$ 30 kg/m$^2$, $\approx$ 23%).

Concerning the PE risks and diverse outcomes, primipaternity women as compared with stable couples had a major higher risk of PE OR 3.39, with a slightly higher risk of early onset PE (EOP, delivery < 34 weeks gestation), OR 3.7 compared with late onset PE (LOP, 34 weeks onward) OR 3.25, p< 0.0001. Primipaternity multiparas had more histories of preceding PE, OR 1.5, preceding birth of an SGA baby, OR 2.2, volunteer abortions, OR 1.67, preceding mis-carriages, OR 1.2, all p< 0.0001. For the current index pregnancy, primipaternity multiparas had more small for gestational age newborns (SGA, birthweight <10th percentile), OR 1.7, and low birthweight (2500g) babies, OR 2.2, p < 0.0001. New paternity multiparas had on average higher birth intervals (of ap. 1.5 years) as compared with stable couples, p< 0.0001.

Table 2 **Mean intervals between pregnancies (in years) and mean maternal ages, stable couples and changing paternity** Systematically, primipaternity multiparas had significant higher intervals of pregnancies as compared with stable couples whatever the ranks of preg-nancies: Approximately 1.5 years for the second to the fourth pregnancy ("intervals second pregnancy" being the interval between baby 1 and 2), and ap. 1 year for pregnancies over the 5th rank. Mean maternal age at all pregnancy ranks were higher in primipaternity multiparas.

Table 3 and Fig 1. The multiple logistic regression model includes 11 items of which the 2 obligatory items having both a regular effect on preeclampsia risk: maternal ages and maternal pre-pregnancy BMI.

It is interesting in logistic models to look first at negative coefficients (meaning a protective effect towards the outcome). Although non-significant, history of volunteer abortions had a tendency to lower the incidence of PE by 13% (coefficient -0.13) and even by 34% for EOP (early onset PE). Smoking during pregnancy also had a tendency to lower the incidence of PE by 16%, effect only due to the 26% decrease concentrated only in LOP (late onset PE), and no protection towards EOP.

The three major independent risk factors, controlling for all the other items in the model were: History of preceding PE: adjusted OR 11.2, p < 0.0001. (Adj OR 17 for EOP), Chronic hypertension: Adj OR 6.45, p < 0.0001. (Adj OR 9.1 for EOP) and primipaternity for the index pregnancy: Adj OR 3.34, p < 0.0001. (3.8 for EOP and 3.1 for LOP). Controlling for the 3 major preceding factors intervals between pregnancies belongs rather to the what we may call "regular physiological aging processes" presenting a regular smooth slope rise: maternal ages coefficient 0.018 (rise of 2% for each increase of 5 years of age), parities 1-2-3-4-5 and over, coef. 0.09 (rise of 9% for each successive pregnancy), interval between pregnancies, coef 0.05. Pre-pregnancy BMI has a coefficient of 0.04 (rise of 4% for each increasing category by 5 kg/m$^2$), effect mainly due to LOP, Adj OR 1.06, Coef 0.06, p< 0.0001.

**Table 1. Maternal characteristics.** Preeclampsia risks. Multiparous, 2$^{nd}$ pregnancy and over.

| Maternal characteristics = > | Controls: Multiparous with same father N = 50,782 (%) | Multiparous with NEW father for the index pregnancy N = 2790 (%) | OR [95% CI] | P value |
|---|---|---|---|---|
| Maternal age (years: mean ± sd) | 30.0 ± 6.03 | 31.2 ± 6.5 | | <.0001 |
| Gravidity ± sd | 3.7 ± 1.9 | 4.3 ± 2.1 | | <.0001 |
| Parity ± sd | 2.0 ± 1.5 | 2.3 ± 1.6 | | <.0001 |
| Adolescents < 18 years | 305 (0.6) | 22 (0.8) | 1.32 [0.9–2.0] | 0.21 |
| Age ≥ 35 years | 12,562 (24.7) | 950 (34.1) | 1.57 [1.45–1.7] | <.0001 |
| Grand multiparas (≥ 5) | 6404 (12.6) | 500 (17.9) | 1.51 [1.4–1.7] | <.0001 |
| Education 10th grade or over | 26,606 (55.0) | 1413 (51.9) | 0.88 [0.82–0.95] | 0.002 |
| Living single | 15,009 (29.8) | 1857 (66.7) | 4.7 [4.4–5.1] | <.0001 |
| BMI kg/m² (mean ± sd) | 25.9 ± 6.5 | 25.8 ± 6.4 | | 0.47 |
| Obesity ≥ 30 kg/m² | 11,289 /48,538 (23.3) | 611/2667 (22.9) | 1.33 [0.89–1.1] | 0.67 |
| Smoking | 6785 (13.4) | 664 (23.8) | 2.0 [1.85–2.2] | <.0001 |

| Preeclampsia risks = > | Controls: Multiparous with same father N = 50,782 (%) | Multiparous with NEW father for the index pregnancy N = 2790 (%) | OR [95% CI] | P value |
|---|---|---|---|---|
| Incidence preeclampsia (%) | 1049 (2.1) | 186 (6.7) | 3.39 [2.9–4.0] | <.0001 |
| Incidence EOP (%) < 34 weeks | 326 (0.7) | 63 (2.4) | 3.7 [2.8–4.9] | <.0001 |
| Incidence LOP (%) ≥ 34 weeks | 723 (1.4) | 123 (4.5) | 3.25 [2.7–4.0] | <.0001 |
| History of elective abortion | 10,040 (19.8) | 813 (29.1) | 1.67 [1.5–1.8] | <.0001 |
| History of miscarriage | 12,143 [23.9] | 764 (27.4) | 1.2 [1.1–1.3] | <.0001 |
| History of previous preeclampsia | 825 (1.6) | 67 (2.4) | 1.5 [1.2–1.9] | <.0001 |
| Previous birth of an SGA newborn | 476 (0.9) | 57 (2.0) | 2.2 [1.7–2.9] | <.0001 |
| SGA < 10th percentile | 4162 (8.3) | 361 [13.3] | 1.7 [1.5–1.9] | <.0001 |
| Low birthweight < 2500g | 2216 (4.7) | 574 (9.6) | 2.2 [2.0–2.4] | <.0001 |
| Intervals between pregnancies (years) - Mean ± SD - Median | 4.3 ± 2.8 / 3.7 | 5.9 ± 3.7 / 5.2 | | <.0001 / <.0001 |

EOP: early onset preeclampsia. LOP: late onset preeclampsia. SGA: small for gestational age (< 10th percentile Reunion curve)

**Table 2. Mean intervals between pregnancies (in years) and mean maternal ages, stable couples and changing paternity.**

| | All women Intervals N = 33,812 | Stable couples Same father Intervals N = 31,452 | Changing father Intervals N = 1858 | Intervals P value Stable & new father | All women Mean ages | Stable couples Same father Mean ages | Changing father Mean ages | Maternal ages P value Stable vs new father |
|---|---|---|---|---|---|---|---|---|
| 2nd pregnancy Mean ± SD | 3.8 ± 2.5 N = 15,981 | 3.7 ± 2.4 N = 15,289 | 5.1 ± 3.0 N = 691 | < 0.0001 | 27.9 ± 5.7 N = 26,091 | 27.9 ± 5.6 N = 25,003 | 28.3 ± 6.2 N = 1086 | 0.04 |
| 3rd pregnancy Mean ± SD | 4.3 ± 2.8 N = 8493 | 4.2 ± 2.7 N = 7996 | 5.8 ± 3.1 N = 496 | < 0.0001 | 30.6 ± 5.6 N = 14,023 | 30.6 ± 5.6 N = 13,268 | 31.5 ± 6.1 N = 754 | < 0.0001 |
| 4th pregnancy Mean ± SD | 4.0 ± 2.7 N = 4117 | 3.9 ± 2.7 N = 3811 | 5.3 ± 3.1 N = 306 | 0.0005 | 32.5 ± 5.5 N = 6558 | 32.4 ± 5.5 N = 6108 | 33.4 ± 5.8 N = 450 | 0.0002 |
| 5th pregnancy Mean ± SD | 3.7 ± 2.5 N = 2129 | 3.6 ±2.5 N = 1961 | 4.6 ± 2.8 N = 168 | 0.04 | 33.5 ± 5.3 N = 3218 | 33.5 ± 5 N = 2979 | 34.2 ± 5.1 N = 238 | 0.04 |
| 6th pregnancy & over Mean ± SD | 2.9 ± 2.9 N = 2592 | 2.9 ± 2.1 N = 2395 | 3.4 ± 2.4 N = 197 | 0.007 | 35.5 ± 4.7 N = 3694 | 35.5 ± 4.7 N = 3424 | 36.0 ± 4.7 N = 262 | 0.09 |

**Table 3. Multiple logistic model with different outcomes Preeclampsia (PE), early onset PE, Late onset PE.** Singleton multiparous N = 53,584.

| | Preeclampsia | | | | | EARLY ONSET PE (< 34 weeks gestation) | | | | LATE ONSET PE (34 weeks gestation onward) | | | |
|---|---|---|---|---|---|---|---|---|---|---|---|---|---|
| | Coeff. | OR | 95% CI | P | Coeff. | OR | 95% CI | P | Coeff. | OR | 95% CI | P | |
| Maternal Age (increment of 5 years age) | 0.018 | **1.02** | [1.0–1.04] | 0.02 | 0.009 | **1.01** | [0.98–1.04] | 0.55 | 0.02 | **1.02** | [1.0–1.04] | 0.02 | |
| Pre-pregnancy BMI (increment of 5 kg/m$^2$) | 0.04 | **1.05** | [0.96–1.01] | < 0.0001 | 0.03 | **1.03** | [1.0–1.05] | 0.007 | 0.06 | **1.06** | [1.04–1.07] | < 0.0001 | |
| Chronic hypertension | 1.86 | **6.45** | [5.0–8.3] | < 0.0001 | 2.2 | **9.1** | [5.9–14] | < 0.0001 | 1.9 | **6.6** | [4.9–8.8] | < 0.0001 | |
| New father for the index pregnancy | 1.20 | **3.34** | [2.7–4.2] | < 0.0001 | 1.35 | **3.85** | [2.5–5.8] | < 0.0001 | 1.12 | **3.08** | {2.4–4.0] | < 0.0001 | |
| Parity: 1-2-3-4-5 and over | 0.09 | **1.09** | [1.02–1.17] | 0.02 | 0.04 | **1.04** | [0.9–1.2] | 0.55 | 0.07 | **1.08** | [0.99–1.16] | 0.06 | |
| History previous preeclampsia | 2.4 | **11.2** | [8.8–14.2] | 0.0007 | 2.84 | **17.2** | [11.8–25.2] | < 0.0001 | 2.26 | **9.6** | [.7.7–12.1] | 0.02 | |
| History of previous SGA newborn | 0.38 | **1.47** | [0.71–3.0] | 0.29 | 0.79 | **2.2** | [0.69–70.1] | 0.17 | 0.19 | **1.2** | [0.5–2.99] | 0.66 | |
| **History of volunteer abortion** | -0.13 | **0.88** | [0.7–1.07] | 0.20 | -0.34 | **0.71** | [0.5–1.8] | 0.11 | -0.05 | **0.95** | [0.76–1.19] | 0.66 | |
| **History of miscarriage** | 0.02 | **1.02** | [0.85–1.2] | 0.80 | -0.01 | **0.98** | [0.7–1.4] | 0.94 | 0.02 | **1.02** | [0.83–1.26] | 0.81 | |
| Interval between pregnancies (years) | 0.05 | **1.05** | [1.02–1.09] | 0.002 | 0.07 | **1.07** | [1.0–1.14] | 0.03 | 0.03 | **1.03** | [0.99–1.07] | 0.07 | |
| Smoking during pregnancy | -0.16 | **0.85** | [0.66–1.09] | 0.20 | -0.005 | **0.99** | [0.63–1.6] | 0.98 | -0.26 | **0.77** | [0.6–1.03] | 0.08 | |

Fig 1 visualizes all the significant adjusted odds ratios with two 2 kinds of risks: those with a low-value (Ajusted OR < 1.1): maternal ages, successive parities, pre-pregnancy BMI and intervals between pregnancies.and the ranking of major risks: history of preeclampsia Adj OR 11, Chronic hypertension AdjOR 6.45 and new paternity Adj OR 3.34.

## Discussion

The results of this large population study with detailed records on the relevant variables clearly demonstrates the presence of two kinds of significant risk factors (Fig 1): Major risk factors: history of PE, chronic hypertension and primipaternity with positive coefficients of respectively 2.4, 1.9 and 1.2. (a risk rise of 24%, 19% and 12% when these risks exist) and significant risk factors with a modest incremental (low coefficients) effect with positive coefficients around 0.018 (maternal ages), 0.09 (successive parities), 0.04 (pre-pregnancy BMIs) and 0.05 (intervals between pregnancies).

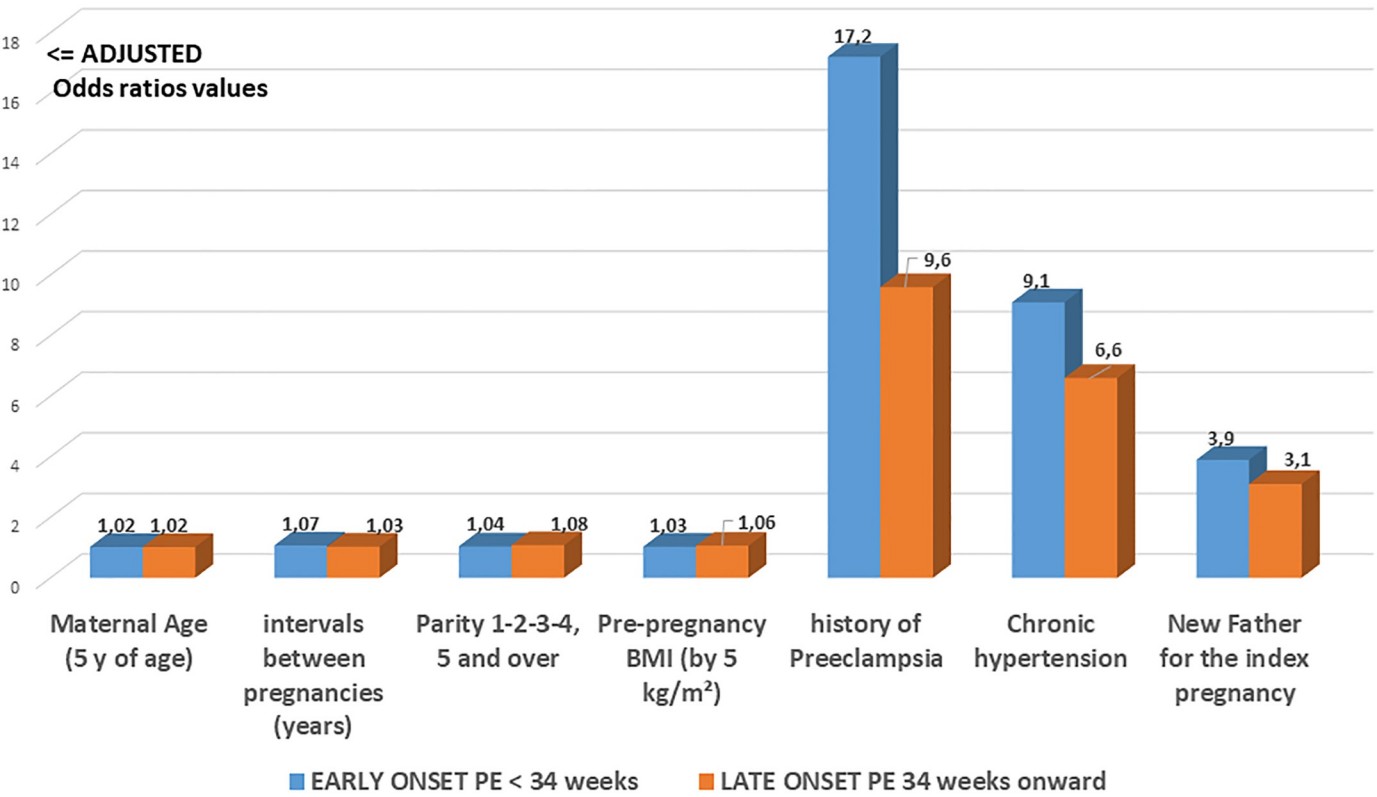

**Fig 1. Significant adjusted odds ratios of several risk factors for early (EOP) and late onset (LOP) preeclampsia.**

Primipaternity multiparous women are in average older than those of stable couples, but also have systematically longer birth intervals compared with controls, see Table 2. A real mixed conundrum of the 2 pivotal competing and concurrent risk factors (paternity and inter-birth intervals). However, in Table 3, using a multiple logistic regression model, all the adjusted odds-ratios giving the true weight of each item, demonstrating the major independent effect of a change in paternity.

That the PE risk may increase by parity, interbirth interval and advanced maternal age has been also clearly detected and written down by Tandberg et al. in 2013 [11], but they did not emphasize that the adjusted OR of these risks were clearly of a much lower magnitude compared with conditions such as chronic hypertension or history of PE. In our model, primipaternity clearly belongs to the cluster of major risks factors after controlling for ages, intervals, BMIs and parities. Intervals between pregnancies have a significant effect, p = 0.002, Table 3 with an adjOR of 1.05, a lower effect than successive pregnancies (adjOR 1.09), pre-pregnancy BMI (adjOR 1.05), but higher than advancing maternal age (adjOR 1.02). These four well-known risks represent a different cluster totally apart with major risk factors such as primigravidty, primipaternity, history of PE or chronic hypertension.

In explaining the the primipaternity paradigm, to date, two major hypotheses (which may be complementary) and are extensively developed in some recent paper [8,16]. Firstly, David

Haig's (check refencing) paternal maternal conflict in pregnancy [17] and secondly, the immunological challenge [18,19] which is also associated with the concept of a necessary long sperm exposure (paternal tissues) in first pregnancies results in a partner specific mucosal tolerance [20]. The haemochorial placenta in primates, and in particular in humans with the deepest invasion represents a scenario where the mother is facing a more or less human-specific major immune challenge, the "fetal hemi-allograft paradox" [19,21] first mentioned by Medawar, prior pregnancies in the same relationship translate in developed "trained immunological memory" in line with well-known epidemiologic findings of lower rates of PE, fetal growth restriction, fetal demise, and low birthweight in subsequent pregnancies. Moreover, we now appreciate that shallow endovascular trophoblast invasion is primarily linked to IUGR (with or without the maternal syndrome of preeclampsia) [22,23].

The strength of our study is, first, the capturing of all perinatal outcomes in our maternity, European standard of care. With 4,300 births per year, the university maternity, level 3, represents 82% of all births in the south of the island. but, as level 3 manages all cases of PE.The data in this large cohort are homogeneous as they were collected in a single center (lower variability) and not based on national birth registers but directly from medical records (avoiding inadequate codes). Second: Reunion island is an overseas department of France in the Indan ocean, and, as such, has the highest per capita gross domestic income of the area (e.g South-Africa, Mozambique, Madagascar, Mauritius island, Seychelles etc. . ..). Furthermore, access to maternity care free of charge as provided by the French healthcare system which combines freedom of medical practice with social security. Hospitals have European standards of care and pregnant have an average of 8 prenatal visits and 4 ultrasounds during their pregnancy.Third, The specificity of our cohort with cultural and social and genetic backgrounds different than other already published studies made in different populations could yield different results. Reunion Island is a French department in the Southern Indian Ocean. The peculiarity of this tropical region lays in the multiethnic origin of inhabitants: Africa and intermixed population (50%), Europe (27%), India (20%) and China (3%). However, we feel this represents a strength of this particular study: this territory witnessed two centuries of slavery until 1848, where official marriages were forbidden for slaves by masters. These communities then reacted by reproducing often with different fathers in successive pregnancies. This scheme remains embedded in this society, where it is not a problem for women to say if a father is a new one or not (well-known pattern which has been described for 4–5 decades by demographers as "Women Family Structures" in the Caribbean's or in American areas where slavery existed).

A weakness of this study is that primipaternity was obviously underestimated as this issue has been added in our database and then prospectively recorded only since 2018. Since then, we have approximately 190 multiparas per year having a new male partner (5.6% of our multiparas). New paternity was recalled during the period 2001–2017 on free commentaries possible in our database. These recalls based on free commentaries are then probably under-represented. We may assume that the retrieved free commentaries on paternity have been biased towards the risk of PE (as primipaternity is known to be associated with this disease [8]. However, we feel that this possible over-representation of preeclamptic pregnancies may be a "good bias" as, controlling for PE and birth intervals in multiparae, primipaternity remains a strong independent risk factor for PE, both for EOP but also for LOP).

## Conclusion

Primipaternity is a markedly stronger risk for PE in multiparous women compared with the modest effect of prolonged birth intervals. Primipaternities in multiparae (immunological

basis) belongs to the major risk factors for PE such as history of preeclampsia, chronic hypertension and multiple pregnancies while prolonged birth intervals belongs to moderate "regular physiological aging processes" such as increasing maternal age, parity or increasing pre-pregnancy BMI.

## Supporting information

**S1 Checklist. CHECK LIST PLOS: Fulfilling of the check list.**
(DOCX)

**S1 File. CALCINTERV22: Calculations of results presented in Table 2.** BOMBINTERV Calculations for a Figure which has been removed from the final manuscript (after advices of reviewers. Figure shown in this file. EXPLORE INTERVALLESDOC: Calculations on maternal ages and BMI. INTERVAL: Calculations of results presented in Table 1. INTERVAL30: Preliminary results number of women by different parities 1-2-3etc. . ..INTERVAL30B: Intervals between pregnancies by different parities 1-2-3etc. . .. comparisons between cases and controls presented also in Table 2. PRIMIPAT24: Complementary calculations of results presented in Table 1. LOGISTIC PAGE 1-2-3: Logistic regression results (Table 3).
(RAR)

## Author Contributions

**Conceptualization:** Pierre-Yves Robillard, Simon Lorrain, Gustaaf Dekker.

**Data curation:** Pierre-Yves Robillard, Phuong Lien Tran.

**Formal analysis:** Pierre-Yves Robillard, Simon Lorrain, Gustaaf Dekker.

**Funding acquisition:** Silvia Iacobelli, Malik Boukerrou.

**Investigation:** Pierre-Yves Robillard, Simon Lorrain, Phuong Lien Tran.

**Methodology:** Pierre-Yves Robillard, Simon Lorrain, Gustaaf Dekker.

**Project administration:** Malik Boukerrou.

**Resources:** Silvia Iacobelli, Malik Boukerrou.

**Software:** Simon Lorrain.

**Supervision:** Pierre-Yves Robillard, Silvia Iacobelli, Marco Scioscia, Gustaaf Dekker.

**Validation:** Pierre-Yves Robillard, Silvia Iacobelli, Simon Lorrain, Francesco Bonsante, Malik Boukerrou, Marco Scioscia.

**Writing – original draft:** Pierre-Yves Robillard.

**Writing – review & editing:** Pierre-Yves Robillard, Francesco Bonsante, Marco Scioscia, Gustaaf Dekker.

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
