## [Decision Letter · Decision Letter 0]

22 Jul 2024

PONE-D-24-23454RELATIVE INFLUENCE OF PRIMIPIPATERNITY AND PROLONGED BIRTH INTERVAL IN MULTIPARAE AS INDEPENDENT RISK FACTORS FOR PREECLAMPSIA;A STUDY OF 33.000 SINGLETON PREGNANCIES IN REUNION ISLANDPLOS ONE

Dear Dr. Robillard,

Thank you for submitting your manuscript to PLOS ONE. After careful consideration, we feel that it has merit but does not fully meet PLOS ONE’s publication criteria as it currently stands. Therefore, we invite you to submit a revised version of the manuscript that addresses the points raised during the review process.

We look forward to receiving your revised manuscript.

Kind regards,

Preenan Pillay

Academic Editor

PLOS ONE

Journal Requirements:

Additional Editor Comments:

The following editorial comments must be addressed:

- Please revise the manuscript to meet the required formatting guidelines presented on our website. https://journals.plos.org/plosone/s/submission-guidelines

- The substance of the work is noted however the authors should refrain from discrediting other authors but rather scientifically contrast the different viewpoints of the argument. The entire manuscript needs to be rewritten to remove this as it presents an ethical non-conformance for PloS One. These types of comments are mainly evident in the introduction and conclusion of the manuscript. One example is as follows: (“The 2002 Skjaerven et al.'s statement: After adjustment for the interval birth, a change of partner is not associated with an increased risk of preeclampsia” severely disorientated the preeclampsia debate in a wrong direction during two decades.).

- The rationale and aim of the study must be rewritten with clarity and in an acceptable scientific format.

- The materials and methods must be written logically and scientifically in alignment with the work and claims made. Segmenting the methods will help in creating scientific flow which will enhance the readability. Importantly the study population size and design is not clear.

- The manuscript must be edited for English, neatness, syntax, grammar and ensure that it is written scientifically.

- The title must be revised to be more concise without being similar to other articles already published and aligned to the actual work done.

- It is suggested that the authors use statistical graphs where possible to clearly represent the results.

- The results section is inconsistent and needs to be rewritten. The results must be clearly specified which must be in alignment with the title and entire rationale of the study.

- The discussion must be revised for coherency and to critically discuss the key findings of the presented study.

- The authors should include a conflict of interest and discloser statement as per the Plos ethical guidelines and manuscript template

The following reviewer comments must be addressed:

Reviewer 1

Overall comments:

Positive Comments:

• Very detailed analysis and interpretation of results

• Novel study in term of cohort and factors studied.

• Large sample size provides more realistic picture of the risk factors

• Wealth of data to write other papers

• Study period of 24 years also gives credibility to the results showing changes over time

Areas to Address:

• The authors do not mention the specificity of the cohort having any bearing on the results. Other studies use different populations hence the disparity between results, different cultural and social and genetic backgrounds will yield different results.

• Study lacks theories or hypothesis as to why the results are such but rather authors explain the findings without explanation.

• Different font sizes and types show different sections written by different people but not carefully synthesized into one coherent paper.

Other specific comments per section:

Introduction

1. References are not consistent i.e. full stop after et al

2. Use of brackets after reference number [8,9] missing a closed bracket

MATERIAL AND METHODS.

3. Word missing...”at the maternity …” ward, department or unit?

4. Elaborate on how gestational hypertension and eclampsia were defined?

5. paragraph 5 after “Definition of exposure and outcomes”: ore-pregnancy should read pre-pregnancy

Results:

6. results need to be neatened up; data is scattered.

7. The narrative on page 15 for the figure has a duplication of the % sign

8. Check all figures containing decimals sone are represented as < 0..0001

Discussion

9. Discussion is very results and statistics-heavy heavy it needs more theory and explanation of the results

Reviewer 2

Comment #1: In the Abstract, Design and Main Outcome Measures should be restructured and described in more details related to PE risk factors, not only stated “… on obstetrical and neonatal risk factors” (?).

Comment #2: The objective of this study should be rephrased, clearly define the “problem” and without parentheses: “The aim of this study was to analyze the problem in our 24-year cohort in Reunion island using our detailed perinatal database where we have an item “changing paternity” by direct inquiry to the women (preeclamptic and controls).

Comment #3: The study period “from January 1st, 2001, to December 31, 2023” is 23 years.

Comment #4: In Results, tables and figures should be clearly formatted and figures with better resolution.

Comment #5: The authors stated “All relevant data are within the manuscript and its Supporting Information files”, but there is no additional data file available.

Comment #6: Excessive self-citation should be avoided and referencing the works from others are encouraged.

Comment #7: The manuscript should be checked carefully by a native for English spelling and grammar.

Reviewers' comments:

Reviewer's Responses to Questions

**Comments to the Author**

1. Is the manuscript technically sound, and do the data support the conclusions?

Reviewer #1: Partly

Reviewer #2: Yes

2. Has the statistical analysis been performed appropriately and rigorously? 

Reviewer #1: Yes

Reviewer #2: Yes

3. Have the authors made all data underlying the findings in their manuscript fully available?

Reviewer #1: No

Reviewer #2: Yes

4. Is the manuscript presented in an intelligible fashion and written in standard English?

Reviewer #1: No

Reviewer #2: No

5. Review Comments to the Author

Reviewer #1: The authors reported the secondary analysis about the two risk-factors in question for development of peeclampsia (PE) among multipara pregnant women: (1) primipaternity, and (2) prolonged birth/pregnancy interval.

Despite its scientific merits, there are some flaws that should addressed, before re-submission for reconsideration.

Comment #1: In the Abstract, Design and Main Outcome Measures should be restructured and described in more details related to PE risk factors, not only stated “… on obstetrical and neonatal risk factors” (?).

Comment #2: The objective of this study should be rephrased, clearly define the “problem” and without parentheses: “The aim of this study was to analyze the problem in our 24-year cohort in Reunion island using our detailed perinatal database where we have an item “changing paternity” by direct inquiry to the women (preeclamptic and controls).

Comment #3: The study period “from January 1st, 2001, to December 31, 2023” is 23 years.

Comment #4: In Results, tables and figures should be clearly formatted and figures with better resolution.

Comment #5: The authors stated “All relevant data are within the manuscript and its Supporting Information files”, but there is no additional data file available.

Comment #6: Excessive self-citation should be avoided and referencing the works from others are encouraged.

Comment #7: The manuscript should be checked carefully by a native for English spelling and grammar.

Reviewer #2: 1. The Manuscript is technically sound, with rigorous statistical techniques and data availability.

2. The manuscript does need major revision for English, neatness, syntax and grammar.

3. major concern is the scarcity of linking results with physiological background and theory.

4. authors fail to hypothesize the rationale behind finding these results and the risk factors.

5. the niche population which could assist in explaining the results are not mentioned. e.g. is it a developing country with poor healthcare standards. lack of education hence seeking multiple fathers etc.

6. attachment uploaded with specific areas of improvement.

6. PLOS authors have the option to publish the peer review history of their article (what does this mean?). If published, this will include your full peer review and any attached files.

Reviewer #1: No

Reviewer #2: No

---

## [Author Response · Author response to Decision Letter 0]

3 Aug 2024

REVIEWERS’ COMMENTS.

1. EDITOR COMMENTS. Answers in green in the text

We have added, page 6 : Ethics approval: This study was conducted in accordance with French legislation. As per new French law applicable to trials involving human subjects (Jardé Act), a specific approval of an ethics committee (comité de protection des personnes- CPP) is not required for this non-interventional study based on retrospective, anonymized data of authorized collections and written patient consent is not needed. Patients and Public involvement. The South-Reunion perinatal database (since 2001) includes 264 items. It is considered as a fully medical database , datasheets are electronically completed solely by midwives, obstetricians and neonatologists. All epidemiological studies are obligatorily performed on anonymized data (French law). As such, there is no direct patient or public involvement.

Additional Editor Comments:

The following editorial comments must be addressed:

- Please revise the manuscript to meet the required formatting guidelines presented on our website. https://journals.plos.org/plosone/s/submission-guidelines

- The substance of the work is noted however the authors should refrain from discrediting other authors but rather scientifically contrast the different viewpoints of the argument. The entire manuscript needs to be rewritten to remove this as it presents an ethical non-conformance for PloS One. These types of comments are mainly evident in the introduction and conclusion of the manuscript. One example is as follows: (“The 2002 Skjaerven et al.'s statement: After adjustment for the interval birth, a change of partner is not associated with an increased risk of preeclampsia” severely disorientated the preeclampsia debate in a wrong direction during two decades.).

We have re-written the INTRODUCTION, deleting 6 lines (essentially after new modification now in dark gree in the text). The modifications beginning at the 3rd paragraph. We have deleted the sentence « was published in the very prestigious NEJM ».

The new text now is :

« The Skjaerven et al.’s NEJM paper had a tremendous impact as it directly disagreed with our preceding publications on the importance of ‘primipaternity’ in 1993 and 1994 (the last one in The Lancet) [4,5]. In those studies we showed that in Guadeloupe (French West Indies) a change of paternity for the index pregnancy, based on direct inquiries with multiparas, was strongly associated with PE [5]. Over the past decades the relative importance of prolonged birth interval versus primipaternity has remained a controversial topic.. From an etiological perspective, this ongoing scientific debate is not a trivial one; evidently, the primipaternity paradigm is in line with the fundamental concept that human placentation may be considered as a “fetal hemigraft” , and as such the classic superficial cytotrophoblast invasion of the spiral particularly in early-onset PE with fetal growth restriction could therefore represent a type of immunological maladaptation of this fetal hemigraft [7-10]. On the other hand, prolonged interval between pregnancies in multiparas as a direct major risk factor for PE appears to be more in line with a kind of vascular maternal problem that increases progressively with time (a kind of “aging approach”). This was also supported by Tanberg et al. (also in Norway) who concluded on a cohort of 500,000 mothers after assisted reproductive technologies (ART, period 1988 to 2009 – so with a ten year overlap of the prior Skjaerven cohort) that the PE risk may increase by parity, interbirth interval and advanced maternal age, but with not with change of father or smoking [11]. In contrast, in 2000 a study published by Li and Wi based on 140,147 women with two consecutive births (Californian birth certificate 1989-1991) [12] among women without PE/eclampsia in the 1st pregnancy, changing partners resulted in a 30% increase in the risk in the subsequent pregnancy compared with those who did not change partner (95% CI: 1.1-1.6). On the other hand, among women with PE in the 1st birth, changing partner resulted in a 30% reduction in the risk of PE in the subsequent pregnancy (95% CI: 0.4- 1.2). Interbirth interval was very unlikely to be a confounder in the Li and Wi study since the authors restricted their population to births that were between 1-3 years apart [12]. Hercus and Dekker in an Australian population studied this problem in 2020 [13] and concluded that “both prolonged birth intervals and primipaternity are independent risk factors for preeclampsia in multigravidae”. 

Notwithstanding the fact that we have previously discussed some concerns regarding the Skjaerven’s study [6-8], it is clear that the relative importance of prolonged birth interval versus primipaternity as PE risk factor in multiparous women still represents an important fundamental research question. 

The aim of this study was to address this fundamental question by a comprehensive analysis of our 23year pregnancy cohort in Reunion island using a detailed high quality perinatal database where we have an item “changing paternity” by direct inquiry to the women (PE and controls).”

 IN THE CONCLUSION we deleted the sentence

“The 2002 Skjaerven et al.'s statement: “After adjustment for the interval birth, a change of partner is not associated with an increased risk of preeclampsia” severely disorientated the preeclampsia debate in a wrong direction during two decades. »

- The rationale and aim of the study must be rewritten with clarity and in an acceptable scientific format.

In the abstract, we modified :

Objectives : To evaluate the relative importance of changing paternity (“primipaternity”) in multiparas versus prolonged birth/pregnancy interval as risk factors for preeclampsia (PE). 

 Main outcome Measures. The aim of this study was to analyze the problem in our population using our detailed perinatal database where we have an item “changing paternity” by direct inquiry to the women (preeclamptic and controls). Comparison of risk factors for PE between multiparae with a new male partner for the index pregnancy versus stable couples.

- The materials and methods must be written logically and scientifically in alignment with the work and claims made. Segmenting the methods will help in creating scientific flow which will enhance the readability. Importantly the study population size and design is not clear.

At the beginning of material and methods we added:

The aim of this study was to evaluate the relative importance of changing paternity (“primipaternity”) in multiparas versus prolonged birth/pregnancy interval as risk factors for preeclampsia (PE) using in our perinatal database the item “changing paternity” by direct inquiry to the women (preeclamptic and controls). Comparison of risk factors for PE between multiparae with a new male partner for the index pregnancy versus stable couples.

Page 5, we wrote :

Definition of exposure and outcomes . During the 23-year period all consecutive singleton pregnancies after 22 weeks gestation have been analysed. For multiparas, women who had changed the male partner for the index pregnancy were considered as cases, those who did not were the controls.

- The manuscript must be edited for English, neatness, syntax, grammar and ensure that it is written scientifically.

Professor Dekker verified entierly the text

- The title must be revised to be more concise without being similar to other articles already published and aligned to the actual work done.

THE NEW TITLE IS NOW :

PRIMIPATERNITY IN MULTIPARAS IS HIGHLY PREDOMINANT AS RISK FACTOR FOR PREECLAMPSIA OVER PROLONGED BIRTH INTERVALS. A STUDY OF 33.000 SINGLETON PREGNANCIES IN REUNION ISLAND

SHORT TITLE :

 PREECLAMPSIA AND MULIPARAS: PRIMIPATERNITY IS A HIGHER RISK FACTOR THAN PROLONGED BIRTH INTERVALS

- It is suggested that the authors use statistical graphs where possible to clearly represent the results.

We feel that Figure 2 : Significant adjusted odds ratios of several risk factors for early (EOP) and late onset (LOP) preeclampsia.

Although unusual gives a visual direct impact for the real impact of the 2 risk factors studied (all results adjusted Odds-ratios)

- The results section is inconsistent and needs to be rewritten. The results must be clearly specified which must be in alignment with the title and entire rationale of the study.

We have made modifications (dark green in the text)

- The discussion must be revised for coherency and to critically discuss the key findings of the presented study.

We have modified a lot the discussion (through the 2 reviewers’ suggestion). We came also from 14 references to 23

- The authors should include a conflict of interest and discloser statement as per the Plos ethical guidelines and manuscript template

At the end of Material and methods we added : « Conflict of interest. The authors report no conflict of interest ».

The following reviewer comments must be addressed:

2. Reviewer 1. Answers in red in the text

Overall comments:

Positive Comments:

• Very detailed analysis and interpretation of results

• Novel study in term of cohort and factors studied.

• Large sample size provides more realistic picture of the risk factors

• Wealth of data to write other papers

• Study period of 24 years also gives credibility to the results showing changes over time

Areas to Address:

• The authors do not mention the specificity of the cohort having any bearing on the results. Other studies use different populations hence the disparity between results, different cultural and social and genetic backgrounds will yield different results.

We have added in strength and limitation, page 14 : « The specificity of our cohort with cultural and social and genetic backgrounds different than other already published studies made in different populations could yield different results. Reunion Island is a French department in the Southern Indian Ocean. The peculiarity of this tropical region lays in the multiethnic origin of inhabitants: Africa and intermixed population (50%), Europe (27%), India (20%) and China (3%). However, we feel it can be a strength for this particular study: this territory witnessed two centuries of slavery until 1848, where official marriages were forbidden for slaves by masters. These communities then reacted by reproduction being often with different fathers in successive pregnancies. This scheme remains embedded in this society, where it is not a problem for women to say if a father is a new one or not (well-known pattern which has been described for 4-5 decades by demographers as “Women Family Structures” in the Carribbeans or in American areas where slavery existed). “

• Study lacks theories or hypothesis as to why the results are such but rather authors explain the findings without explanation.

We have added, page 14 a long paragraph explaining the biological plausibility of the « primipaternity concept » : « What is the biologic explanation for the primipaternity paradigm ? To date, two major hypotheses (which may be complementary) and are extensively developed in some recent paper [8, 16]: First, David Haig’s paternal-maternal conflict in every pregnancy [17], second, the immunological one [18,19] also associated with the concept of a necessary long sperm exposure (paternal tissues) in first pregnancies results in a partner specific mucosal tolerance [20]. Immunology of reproduction has made giant leaps during the last two decades: The haemochorial placenta in primates, and in particular in humans with the deepest invasion represents a scenario where the mother is facing a more or less human-specific major immune challenge, the “fetal hemi-allograft paradox ” [19] first mentioned by Medawar, Prior pregnancies in the same relationship translate in developped ta “trained immunological memory” in line with well known epidemiologic findings of lower rates o f of PE , fetal growth restriction, fetal demise, and low birthweight in subsequent pregnancies. Also, we now appreciate that shallow endovascular trophoblast invasion is primarily linked to IUGR (with or without the maternal syndrome of preeclampsia) [22, 23].

• Different font sizes and types show different sections written by different people but not carefully synthesized into one coherent paper. Corrected

Other specific comments per section:

Introduction

1. References are not consistent i.e. full stop after et al. This reference has been deleted (critic on self-citation)

2. Use of brackets after reference number [8,9] missing a closed bracket

corrected

MATERIAL AND METHODS.

3. Word missing...”at the maternity …” ward, department or unit?

Corrected : department

4. Elaborate on how gestational hypertension and eclampsia were defined?

We have modified page 5 : ”Preeclampsia was defined according to the World Health Organization recommendations [14] and the International Society for the study of Hypertension in Pregnancy ) relatively to the guidelines in force at the year of pregnancy. [15] as the new onset of hypertension (BP ≥140 mmHg systolic or ≥90 mm Hg diastolic) at or after 20 weeks’ gestation and substantial proteinuria (> 0.3 g/24 hours). Early onset preeclampsia was defined as preeclampsia that developed before 34 weeks of gestation.”

5. paragraph 5 after “Definition of exposure and outcomes”: ore-pregnancy should read pre-pregnancy. Corrected

Results:

6. results need to be neatened up; data is scattered. We divided the results to be connected with the different Tables as it is recommended by PLOS (Tables embeded inside the text)

7. The narrative on page 15 for the figure has a duplication of the % sign. Corrected thank you

8. Check all figures containing decimals sone are represented as < 0..0001. Corrected thank you

9 Discussion. Discussion is very results and statistics-heavy heavy it needs more theory and explanation of the results. Paragraph added page 14, pleas see above

10. The authors stated “All relevant data are within the manuscript and its Supporting Information files”, but there is no additional data file available.

Reviewer 2. Answers in blue in the text

Comment #1: In the Abstract, Design and Main Outcome Measures should be restructured and described in more details related to PE risk factors, not only stated “… on obstetrical and neonatal risk factors” (?).

We have precised : « obstetrical and neonatal risk factors (156 items in total, of which, chronic or gestational hypertension, proteinuria, HELLP syndrome).”

Comment #2: The objective of this study should be rephrased, clearly define the “problem” and without parentheses: “The aim of this study was to analyze the problem in our 24-year cohort in Reunion island using our detailed perinatal database where we have an item “changing paternity” by direct inquiry to the women (preeclamptic and controls).

The sentence is now : »The aim of this study was to analyze the problem in our using our detailed perinatal database where we have an item “changing paternity” by direct inquiry to the women (preeclamptic and controls). Comparison of risk factors for PE between multiparae with a new male partner for the index pregnancy versus stable couples.”

Comment #3: The study period “from January 1st, 2001, to December 31, 2023” is 23 years. Corrected thank you

Comment #4: In Results, tables and figures should be clearly formatted and figures with better resolution.

Comment #5: The aut

---

## [Decision Letter · Decision Letter 1]

12 Aug 2024

PONE-D-24-23454R1PRIMIPATERNITY IN MULTIPARAS IS HIGHLY PREDOMINANT AS RISK FACTOR FOR PREECLAMPSIA OVER PROLONGED BIRTH INTERVALS. A STUDY OF 33.000 SINGLETON PREGNANCIES IN REUNION ISLANDPLOS ONE

Dear Dr. Robillard,

Thank you for submitting your manuscript to PLOS ONE. After careful consideration, we feel that it has merit but does not fully meet PLOS ONE’s publication criteria as it currently stands. Therefore, we invite you to submit a revised version of the manuscript that addresses the points raised during the review process.

Please kindly address all recommendations effectively and ensure that you follow all scientific standards in disseminating your findings.==============================

We look forward to receiving your revised manuscript.

Kind regards,

Preenan Pillay

Academic Editor

PLOS ONE

Journal Requirements:

Additional Editor Comments:

Thank you for your revisions. We however believe that there are some minor revisions that need to be made before we consider your manuscript for publication. They are as follows:

Editorial Comments:

General:

- Inclusion of more scientific information in the methods section as indicated.

- Revision of statements for consistency and coherency

- Correct English and grammar

- Correct statements to a more scientific format

-Check refencing consistency according to PlOs One policies

Title suggestion:

Primipaternity in Multiparas as a Predominantly High Risk Factor for Preeclampsia Over prolonged birth intervals: A study of Singleton Pregnancies in Reunion Island

Abstract:

Segment Abstract according to the following sections

- Introduction

- Methods – Mention briefly the actual methods used. There is mention of assessment but not what the actual assessment is about.

- Results

- Conclusion and Relevance

Introduction:

I recommend that the authors change the following paragraphs:

Interestingly Trogstadt et al (with Skjaerven as one of the co-authors) looked at the effect of prior miscarriage/abortions (< 22 week’s gestation) in nulliparous women (Norwegian MoBa cohort) [3] and showed that prior abortion with the same partner reduced the risk for PE but not in women with new paternity pregnancies and the authors concluded that normal pregnancies interrupted at early gestation may induce immunological changes that reduce the risk of preeclampsia in a subsequent pregnancy.

The Skjaerven et al.’s NEJM paper had a tremendous impact as it directly disagreed with our preceding publications on the importance of ‘primipaternity’ in 1993 and 1994 (the last one in The Lancet) [4,5].

To:

Interestingly, Trogstadt et al. investigated the effect of prior miscarriage/abortions (< 22 week’s gestation) in nulliparous women within a Norwegian MoBa cohort [3]. This study identified that prior to abortion with the same partner a reduced risk of PE was observed however, not in women with new paternity pregnancies. These findings suggested that normal pregnancies interrupted at early gestation may induce immunological changes that reduce the risk of preeclampsia in subsequent pregnancies.

Importantly, a study by Skjaerven et al.’s [add reference] had tremendous impact as it directly contrasted our preceding publications on the importance of ‘primipaternity’ [4,5].

Methods:

Add more details on the following methods and parameters applied per the software package used since multiple software platforms were used:

‘Epidemiological data have been recorded and analysed with the software EPI-INFO 7.1.5 (2008, CDC Atlanta, OMS), EPIDATA 3.0 and EPIDATA Analysis V2.2.2.183. Denmar’

I suggest specifying the package used for each statistical test done and also adding more detail on the other tests done using the software.

Results:

- Intext references to tables and figures should be included and must be consistent.

- Authors must adjust figure to a 2D format the 3D format is not for scientific publication as a publication must be as clear as possible to read.

- Authors must not use bullets to disseminate their findings and must write their findings in a clear scientific manner for coherency, the bullet points are okay for power point presentations but not for publication purposes.

Discussion

- Authors must not use bullets to discuss their work and must write their findings in a clear scientific manner for coherency, the bullet points are okay for power point presentations but not for publication purposes.

- I recommend the authors change the following paragraph:

What is the biologic explanation for the primipaternity paradigm ? To date, two major hypotheses (which may be complementary) and are extensively developed in some recent paper [8, 16]: First, David Haig’s paternal maternal conflict in every pregnancy [17], second, the immunological one [18,19] also associated with the concept of a necessary long sperm exposure (paternal tissues) in first pregnancies results in a partner specific mucosal tolerance [20]. Immunology of reproduction has made giant leaps during the last two decades: The haemochorial placenta in primates, and in particular in humans with the deepest invasion represents a scenario where the mother is facing a more or less human-specific major immune challenge, the “fetal hemi-allograft paradox ” [19] first mentioned by Medawar, Prior pregnancies in the same relationship translate in developped ta “trained immunological memory” in line with well known epidemiologic findings of lower rates o f of PE , fetal growth restriction, fetal demise, and low birthweight in subsequent pregnancies. Also, we now appreciate that shallow endovascular trophoblast invasion is primarily linked to IUGR (with or without the maternal syndrome of preeclampsia) [22, 23]

TO

In explaining the the primipaternity paradigm, to date, two major hypotheses (which may be complementary) and are extensively developed in some recent paper [8, 16]. Firstly, David Haig’s (check refencing) paternal maternal conflict in pregnancy [17] and secondly, the immunological challenge [18,19] which is also associated with the concept of a necessary long sperm exposure (paternal tissues) in first pregnancies results in a partner specific mucosal tolerance [20]. The haemochorial placenta in primates, and in particular in humans with the deepest invasion represents a scenario where the mother is facing a more or less human-specific major immune challenge, the “fetal hemi-allograft paradox ” [19] first mentioned by Medawar, prior pregnancies in the same relationship translate in developed “trained immunological memory” in line with well-known epidemiologic findings of lower rates of PE , fetal growth restriction, fetal demise, and low birthweight in subsequent pregnancies. Moreover, we now appreciate that shallow endovascular trophoblast invasion is primarily linked to IUGR (with or without the maternal syndrome of preeclampsia) [22, 23]

- Remove etc from the following sentence and change further to furthermore: (e.g South-Africa, Mozambique, Madagascar, Mauritius island, Seychelles etc….). Further,

Reviewer Recommendations:

Comment #1: At the end of Materials & methods, there is the section “Fundings: The author(s) received no specific funding for this work. Conflict of interest. The authors report no conflict of interest”, this should be removed and inserted at the appropriate position.

Comment #2: In Results, tables and figures should be clearly formatted and figures with better resolution

Comment #3: Many punctuation errors should be corrected.

Reviewers' comments:

Reviewer's Responses to Questions

**Comments to the Author**

1. If the authors have adequately addressed your comments raised in a previous round of review and you feel that this manuscript is now acceptable for publication, you may indicate that here to bypass the “Comments to the Author” section, enter your conflict of interest statement in the “Confidential to Editor” section, and submit your "Accept" recommendation.

Reviewer #1: All comments have been addressed

2. Is the manuscript technically sound, and do the data support the conclusions?

Reviewer #1: Yes

3. Has the statistical analysis been performed appropriately and rigorously? 

Reviewer #1: Yes

4. Have the authors made all data underlying the findings in their manuscript fully available?

Reviewer #1: Yes

5. Is the manuscript presented in an intelligible fashion and written in standard English?

Reviewer #1: No

6. Review Comments to the Author

Reviewer #1: PONE-D-24-23454R1

RELATIVE INFLUENCE OF PRIMIPIPATERNITY AND PROLONGED BIRTH INTERVAL INMULTIPARAE AS INDEPENDENT RISK FACTORS FOR PREECLAMPSIA;A STUDY OF 33.000 SINGLETON PREGNANCIES IN REUNION ISLAND

changed into

PRIMIPATERNITY IN MULTIPARAS IS HIGHLY PREDOMINANT AS RISK FACTOR FOR PREECLAMPSIA OVER PROLONGED BIRTH INTERVALS. A STUDY OF 33.000 SINGLETON PREGNANCIES IN REUNION ISLAND

Most of comments and suggestions raised within the original manuscript have been solved in the revised version R1. The manuscript needs some minor revisions before being considered for publication.

Comment #1: At the end of Materials & methods, there is the section “Fundings: The author(s) received no specific funding for this work. Conflict of interest. The authors report no conflict of interest”, this should be removed and inserted at the appropriate position.

Comment #2: In Results, tables and figures should be clearly formatted and figures with better resolution

Comment #3: Many punctuation errors should be corrected.

7. PLOS authors have the option to publish the peer review history of their article (what does this mean?). If published, this will include your full peer review and any attached files.

Reviewer #1: No

---

## [Author Response · Author response to Decision Letter 1]

20 Aug 2024

1. Journal requirements

We have verified the references which are in the format such as :

Kho EM, McCowan LM, North RA, Roberts CT, Chan E, Black MA, Taylor RS, Dekker GA; SCOPE Consortium. Duration of sexual relationship and its effect on preeclampsia and small for gestational age perinatal outcome. J Reprod Immunol. 2009 Oct;82(1):66-73. doi: 10.1016/j.jri.2009.04.011. Epub 2009 Aug 12. PMID: 19679359.

2. Editorial Comments: ANSWER IN DARK GREEN IN THE TEXT

Additional Editor Comments:

Thank you for your revisions. We however believe that there are some minor revisions that need to be made before we consider your manuscript for publication. They are as follows:

Thank you for these very postive critics which improve the quality f the text. WE HAVE REMOVED FIGURE 1 WHICH ALSO AS YOU SAY « are okay for power point presentations but not for publication purposes. ». Therefore, former Figure 2 becomes Figure 1

General:

- Inclusion of more scientific information in the methods section as indicated.

- Revision of statements for consistency and coherency

- Correct English and grammar

- Correct statements to a more scientific format

-Check refencing consistency according to PlOs One policies

Title suggestion:

Primipaternity in Multiparas as a Predominantly High Risk Factor for Preeclampsia Over prolonged birth intervals: A study of Singleton Pregnancies in Reunion Island

The new title is now (as also sggested by reviewer 1) is :

PRIMIPATERNITY IN MULTIPARAS AS A PREDOMINANT HIGH RISK FACTOR FOR PREECLAMPSIA OVER PROLONGED BIRTH INTERVALS : A STUDY OF 33,000 SINGLETON PREGNANCIES IN REUNION ISLAND

Abstract:

Segment Abstract according to the following sections

- Introduction

- Methods – Mention briefly the actual methods used. There is mention of assessment but not what the actual assessment is about.

We have modified :

Objectives : To evaluate the relative importance of changing paternity (“primipaternity”, direct inquiry with patients) in multiparas versus prolonged birth/pregnancy interval as risk factors for preeclampsia (PE) by a logistic regression model comparing the adjusted odds ratios of both exposures.

- Results

- Conclusion and Relevance

Conclusions. Primipaternities in multiparae belongs to the major risk factors such as history of preeclampsia, chronic hypertension, multiple pregnancies while prolonged birth intervals belongs to moderate “physiological or aging influence” such as increasing maternal age, parity or increasing pre-pregnancy BMI. 

Introduction:

I recommend that the authors change the following paragraphs:

Interestingly Trogstadt et al (with Skjaerven as one of the co-authors) looked at the effect of prior miscarriage/abortions (< 22 week’s gestation) in nulliparous women (Norwegian MoBa cohort) [3] and showed that prior abortion with the same partner reduced the risk for PE but not in women with new paternity pregnancies and the authors concluded that normal pregnancies interrupted at early gestation may induce immunological changes that reduce the risk of preeclampsia in a subsequent pregnancy.

The Skjaerven et al.’s NEJM paper had a tremendous impact as it directly disagreed with our preceding publications on the importance of ‘primipaternity’ in 1993 and 1994 (the last one in The Lancet) [4,5].

To:

Interestingly, Trogstadt et al. investigated the effect of prior miscarriage/abortions (< 22 week’s gestation) in nulliparous women within a Norwegian MoBa cohort [3]. This study identified that prior to abortion with the same partner a reduced risk of PE was observed however, not in women with new paternity pregnancies. These findings suggested that normal pregnancies interrupted at early gestation may induce immunological changes that reduce the risk of preeclampsia in subsequent pregnancies.

Importantly, a study by Skjaerven et al.’s [add reference] had tremendous impact as it directly contrasted our preceding publications on the importance of ‘primipaternity’ [4,5].

NEW TEXTMODIFIED : Interestingly, Trogstadt et al. investigated the effect of prior miscarriage/abortions (< 22 week’s gestation) in nulliparous women within a Norwegian MoBa cohort [3]. This study identified that prior to abortion with the same partner a reduced risk of PE was observed however, not in women with new paternity pregnancies. These findings suggested that normal pregnancies interrupted at early gestation may induce immunological changes that reduce the risk of preeclampsia in subsequent pregnancies. Importantly, a study by Skjaerven et al.’s [1] had tremendous impact as it directly contrasted our preceding publications on the importance of ‘primipaternity’ [4,5]. 

Methods:

Add more details on the following methods and parameters applied per the software package used since multiple software platforms were used:

‘Epidemiological data have been recorded and analysed with the software EPI-INFO 7.1.5 (2008, CDC Atlanta, OMS), EPIDATA 3.0 and EPIDATA Analysis V2.2.2.183. Denmar’

We have added : .EPIDATA allowing the adaptation for WINDOWS 10 of the former EPI-INFO (MS DOS) in complete cooperation with CDC Atlanta. All calculations were made then with and EPIDATA Analysis V2.2.2.183. Denmark.

I suggest specifying the package used for each statistical test done and also adding more detail on the other tests done using the software.

All calculations were made then with and EPIDATA Analysis V2.2.2.183. Denmark.

Results:

- Intext references to tables and figures should be included and must be consistent. DONE

- Authors must adjust figure to a 2D format the 3D format is not for scientific publication as a publication must be as clear as possible to read.

FIGURE 1 (FORMER FIGURE 2) IS NOW IN 2D FORMAT

- Authors must not use bullets to disseminate their findings and must write their findings in a clear scientific manner for coherency, the bullet points are okay for power point presentations but not for publication purposes.

 WE HAVE DELETED THE BULLETS AS WELL IN THE RESULTS AS IN THE DISCUSSION

Discussion

- Authors must not use bullets to discuss their work and must write their findings in a clear scientific manner for coherency, the bullet points are okay for power point presentations but not for publication purposes.

 WE HAVE DELETED THE BULLETS AS WELL IN THE RESULTS AS IN THE DISCUSSION

- I recommend the authors change the following paragraph:

What is the biologic explanation for the primipaternity paradigm ? To date, two major hypotheses (which may be complementary) and are extensively developed in some recent paper [8, 16]: First, David Haig’s paternal maternal conflict in every pregnancy [17], second, the immunological one [18,19] also associated with the concept of a necessary long sperm exposure (paternal tissues) in first pregnancies results in a partner specific mucosal tolerance [20]. Immunology of reproduction has made giant leaps during the last two decades: The haemochorial placenta in primates, and in particular in humans with the deepest invasion represents a scenario where the mother is facing a more or less human-specific major immune challenge, the “fetal hemi-allograft paradox ” [19] first mentioned by Medawar, Prior pregnancies in the same relationship translate in developped ta “trained immunological memory” in line with well known epidemiologic findings of lower rates o f of PE , fetal growth restriction, fetal demise, and low birthweight in subsequent pregnancies. Also, we now appreciate that shallow endovascular trophoblast invasion is primarily linked to IUGR (with or without the maternal syndrome of preeclampsia) [22, 23]

TO

In explaining the the primipaternity paradigm, to date, two major hypotheses (which may be complementary) and are extensively developed in some recent paper [8, 16]. Firstly, David Haig’s (check refencing) paternal maternal conflict in pregnancy [17] and secondly, the immunological challenge [18,19] which is also associated with the concept of a necessary long sperm exposure (paternal tissues) in first pregnancies results in a partner specific mucosal tolerance [20]. The haemochorial placenta in primates, and in particular in humans with the deepest invasion represents a scenario where the mother is facing a more or less human-specific major immune challenge, the “fetal hemi-allograft paradox ” [19] first mentioned by Medawar, prior pregnancies in the same relationship translate in developed “trained immunological memory” in line with well-known epidemiologic findings of lower rates of PE , fetal growth restriction, fetal demise, and low birthweight in subsequent pregnancies. Moreover, we now appreciate that shallow endovascular trophoblast invasion is primarily linked to IUGR (with or without the maternal syndrome of preeclampsia) [22, 23]

 THE NEW PARAGRAPH IS NOW :

In explaining the the primipaternity paradigm, to date, two major hypotheses (which may be complementary) and are extensively developed in some recent paper [8, 16]. Firstly, David Haig’s (check refencing) paternal maternal conflict in pregnancy [17] and secondly, the immunological challenge [18,19] which is also associated with the concept of a necessary long sperm exposure (paternal tissues) in first pregnancies results in a partner specific mucosal tolerance [20]. The haemochorial placenta in primates, and in particular in humans with the deepest invasion represents a scenario where the mother is facing a more or less human-specific major immune challenge, the “fetal hemi-allograft paradox ” [19] first mentioned by Medawar, prior pregnancies in the same relationship translate in developed “trained immunological memory” in line with well-known epidemiologic findings of lower rates of PE , fetal growth restriction, fetal demise, and low birthweight in subsequent pregnancies. Moreover, we now appreciate that shallow endovascular trophoblast invasion is primarily linked to IUGR (with or without the maternal syndrome of preeclampsia) [22, 23]

- Remove etc from the following sentence and change further to furthermore: (e.g South-Africa, Mozambique, Madagascar, Mauritius island, Seychelles etc….). Further,

DONE: Furthermore

3. Reviewer Recommendations: ANSWERS IN RED IN THE TEXT

Comment #1: At the end of Materials & methods, there is the section “Fundings: The author(s) received no specific funding for this work. Conflict of interest. The authors report no conflict of interest”, this should be removed and inserted at the appropriate position.

These have been put in final disclosure

Comment #2: In Results, tables and figures should be clearly formatted and figures with better resolution

Comment #3: Many punctuation errors should be corrected. Done

Reviewers' comments:

Reviewer's Responses to Questions

Comments to the Author

1. If the authors have adequately addressed your comments raised in a previous round of review and you feel that this manuscript is now acceptable for publication, you may indicate that here to bypass the “Comments to the Author” section, enter your conflict of interest statement in the “Confidential to Editor” section, and submit your "Accept" recommendation.

Reviewer #1: All comments have been addressed

2. Is the manuscript technically sound, and do the data support the conclusions?

Reviewer #1: Yes

3. Has the statistical analysis been performed appropriately and rigorously? 

Reviewer #1: Yes

4. Have the authors made all data underlying the findings in their manuscript fully available?

Reviewer #1: Yes

5. Is the manuscript presented in an intelligible fashion and written in standard English?

Reviewer #1: No Professor Dekker revised the English language

6. Review Comments to the Author

Reviewer #1: PONE-D-24-23454R1

RELATIVE INFLUENCE OF PRIMIPIPATERNITY AND PROLONGED BIRTH INTERVAL INMULTIPARAE AS INDEPENDENT RISK FACTORS FOR PREECLAMPSIA;A STUDY OF 33.000 SINGLETON PREGNANCIES IN REUNION ISLAND

changed into

PRIMIPATERNITY IN MULTIPARAS IS HIGHLY PREDOMINANT AS RISK FACTOR FOR PREECLAMPSIA OVER PROLONGED BIRTH INTERVALS. A STUDY OF 33.000 SINGLETON PREGNANCIES IN REUNION ISLAND

The new title (as suggested also by the Editor) is :

PRIMIPATERNITY IN MULTIPARAS AS A PREDOMINANT HIGH RISK FACTOR FOR PREECLAMPSIA OVER PROLONGED BIRTH INTERVALS : A STUDY OF 33,000 SINGLETON PREGNANCIES IN REUNION ISLAND

Most of comments and suggestions raised within the original manuscript have been solved in the revised version R1. The manuscript needs some minor revisions before being considered for publication.

Comment #1: At the end of Materials & methods, there is the section “Fundings: The author(s) received no specific funding for this work. Conflict of interest. The authors report no conflict of interest”, this should be removed and inserted at the appropriate position.

These have been put in final disclosure

Comment #2: In Results, tables and figures should be clearly formatted and figures with better resolution

Figure 1 (former Figure 2) has been reformatted .

Former Figure 1 has been removed

---

## [Editor Report · Decision Letter 2]

28 Aug 2024

PRIMIPATERNITY IN MULTIPARAS AS A PREDOMINANT HIGH RISK FACTOR FOR PREECLAMPSIA OVER PROLONGED BIRTH INTERVALS : A STUDY OF 33,000 SINGLETON PREGNANCIES IN REUNION ISLAND

PONE-D-24-23454R2

Dear Dr. Robillard,

We’re pleased to inform you that your manuscript has been judged scientifically suitable for publication and will be formally accepted for publication once it meets all outstanding technical requirements.

Kind regards,

Preenan Pillay

Academic Editor

PLOS ONE

Additional Editor Comments (optional):

Good day,

Thank you for your revision.

I recommend the submitted manuscript be accepted for publication with the following compulsory amendments:

- Formatting

- English language editing including grammar checks.

- The graphs should not be in 3D and must conform to the graphing standards of PloS One.

- The tables must be formatted to the requirements of PloS One.

- A graphical abstract is recommended but not compulsory.

It is recommended that you kindly conduct these changes in accordance with the PloS one publication guidelines.

Kind Regards

---

## [Editor Report · Acceptance letter]

18 Oct 2024

PONE-D-24-23454R2 

PLOS ONE

Dear Dr. Robillard, 

I'm pleased to inform you that your manuscript has been deemed suitable for publication in PLOS ONE. Congratulations! Your manuscript is now being handed over to our production team.

Kind regards, 

on behalf of

Prof Preenan Pillay 

Academic Editor

PLOS ONE